# Technical note: On comparing greenhouse gas emission metrics

Ian Enting[1] and Nathan Clisby[2]

[1]CSIRO Climate Science Centre, Oceans and Atmosphere, Aspendale, Vic, Australia
[2]Department of Mathematics, Swinburne University of Technology, PO Box 218, Hawthorn Vic, 3122, Australia

**Correspondence:** Ian Enting (ian.g.enting@gmail.com)

**Abstract.** Many metrics for comparing greenhouse gas emissions can be expressed as an instantaneous Global Warming Potential multiplied by the ratio of airborne fractions calculated in various ways. The Forcing Equivalent Index (FEI) provides a specification for equal radiative forcing at all times at the expense of generally precluding point by point equivalence over time. The FEI can be expressed in terms of asymptotic airborne fractions for exponentially growing emissions. This provides a reference against which other metrics can be compared.

Four other equivalence metrics are evaluated in terms of how closely they match the timescale dependence of FEI, with methane, referenced to carbon dioxide, used as an example. The 100-year Global Warming Potential over-estimates the long-term role of methane while metrics based on rates of change over-estimate the short-term contribution. A recently-proposed metric, based on differences between methane emissions 20 years apart, provides a good compromise. Analysis of the timescale dependence of metrics, expressed as Laplace transforms, leads to an alternative metric that gives closer agreement with FEI at the expense of considering methane over longer time periods.

The short-term behaviour, which is important when metrics are used for emissions trading, is illustrated with simple examples for the four metrics.

## 1 Introduction

Anthropogenic contributions to global climate change come from a range of greenhouse gases. Comparisons between them have been facilitated by defining emission-equivalence relations (which we denote by $\equiv$), usually using $CO_2$ as a reference.

The climatic influence of greenhouse gases is commonly represented in terms of radiative forcing, $F$, expressed in terms of $M_X$, the atmospheric mass of gas X, with the effect of small perturbations linearised as

$$\Delta F = a_X \Delta M_X \tag{1}$$

where $a_X$ is the radiative efficiency in mass units: the amount of change in radiative forcing per unit mass increase for constituent $X$ in the atmosphere (Myhre et al., 2013).

Equivalence relations between sources of greenhouse gases are complicated because various gases are lost from the atmosphere on a range of different timescales. This behaviour is often represented using linear response functions, where the response function, $R_X(t)$, represents the proportion of $\Delta S_X$, the perturbation in emissions of constituent X, that remains in the atmosphere after time $t$. Thus the mass perturbation, $\Delta M_X$, is given as a convolution integral:

$$\Delta M_X(t) = \int_0^t R_X(t - t') \, \Delta S_X(t') \, dt' \tag{2}$$

The outline of this note is as follows. In Section 2 we show how the prescription by Wigley (1998), which gives exact equivalence in radiative forcing between different time histories of emissions, may be elegantly expressed in terms of Laplace transforms. In Section 3, we adapt this representation to other metrics of emission equivalence, and use it as inspiration for a new metric with a single adjustable parameter which accurately approximates equivalence in radiative forcing over timescales from decades to multiple centuries. In Section 4, we compare the different metrics in the time domain, and Section 5 discusses some of the mathematical characteristics that may bear on the political acceptance of alternative specifications of emission equivalence. We conclude in Section 6. An appendix expresses the metrics in terms of frequency response and a second appendix lists the notation. Enting and Clisby (2021) references the archived R code used in this paper.

## 2 Metrics: FEI

Wigley (1998) defined an equivalence between emission histories, termed the Forcing Equivalent Index (FEI). Two emission histories are FEI-equivalent if they lead to equivalent forcing at all times. In most cases, this requirement precludes point-by-point emission equivalence at all times.

Equivalent radiative forcing over all time from perturbations $\Delta S_X$ and $\Delta S_Y$ in the emissions of gases X and Y requires:

$$a_Y \int_0^t R_Y(t - t') \, \Delta S_Y(t') \, dt' = a_X \int_0^t R_X(t - t') \, \Delta S_X(t') \, dt' \qquad \text{for all } t \tag{3}$$

as the condition for

$$\Delta S_Y(t) \underset{\text{FEI}}{\equiv} \Delta S_X(t) \tag{4}$$

Subject to the conditions of linearity, this equivalence defines exact equality of radiative forcing. However it is an equivalence for emission profiles and not for instantaneous values.

A special case of FEI-equivalence (see for example Enting, 2018) is when $\Delta S_X$ and $\Delta S_Y$ both grow exponentially, with growth rate $\alpha$ and amplitudes $c_X$ and $c_Y$ at $t = 0$. Exponential growth has

$$\Delta M_X(t) = \int_{-\infty}^t R_X(t - t') \, c_X \exp(\alpha t') \, dt' = c_X \exp(\alpha t) \int_0^\infty R_X(t'') \exp(-\alpha t'') \, dt'' \tag{5}$$

The integral on the right is $\tilde{R}_X(p)$, the Laplace transform of $R_X(t)$, evaluated at $p = \alpha$. Thus for FEI equivalence of emissions growing exponentially at rate $\alpha$ one has

$$c_X = \frac{a_Y \tilde{R}_Y(\alpha)}{a_X \tilde{R}_X(\alpha)} c_Y \tag{6}$$

Interpreting these relations in terms of Laplace transforms can help clarify the different forms of equivalence metrics in the general case. More generally, for arbitrary emission perturbations the condition for FEI-equivalence is defined by the Laplace transform of (3):

$$a_Y \Delta \tilde{S}_Y(p) \tilde{R}_Y(p) = a_X \Delta \tilde{S}_X(p) \tilde{R}_X(p) \tag{7}$$

giving

$$\Delta \tilde{S}_X(p) \underset{\text{FEI}}{\equiv} \frac{a_Y}{a_X} \frac{\tilde{R}_Y(p)}{\tilde{R}_X(p)} \Delta \tilde{S}_Y(p) = \frac{a_Y}{a_X} \tilde{\Psi}_{\text{FEI}}(p) \Delta \tilde{S}_Y(p) \tag{8}$$

In this expression $\tilde{\Psi}_{\text{FEI}}(p) = \tilde{R}_Y(p)/\tilde{R}_X(p)$ is the Laplace transform of an integro-differential operator that, in the time domain, acts on $\Delta S_Y(t)$. Differentiation of (5) shows that, for exponentially growing emissions, the asymptotic airborne fraction of a gas X is $\alpha \tilde{R}_X(\alpha)$ (e.g. Enting, 1990) and so the FEI curve can be defined as the ratio of asymptotic airborne fractions for growth rate $p$.

The plot in Figure 1 describes the specific case of methane, $CH_4$, referenced to carbon dioxide, $CO_2$. The solid line, denoted FEI, can be interpreted in several different, but mathematically equivalent, ways:

- it is the ratio of asymptotic airborne fractions for exponential growth, shown as a function of growth rate;

- it gives the ratio that leads to FEI-equivalence in the special case of exponentially growing emissions;

- it is the Laplace transform of an operator $\Psi_{\text{FEI}}$ that acts on methane emission functions to produce FEI-equivalent $CO_2$ emissions.

In these last two cases, the FEI-equivalence is achieved by scaling by $a_{CH4}/a_{CO2}$.

## 3 Comparison of metrics

The examples given here compare four different metrics, again for the case of $CH_4$ referenced to $CO_2$, benchmarking them against FEI. A general linear, time-invariant equivalence relation can be defined by

$$a_{CO2} \Delta \tilde{S}_{CO2\text{-eq}}(p) = a_{CH4} \tilde{\Psi}(p) \Delta \tilde{S}_{CH4}(p) \tag{9}$$

In the time domain, such a metric can be regarded as a process that extracts, from the history of $CH_4$ emissions, an 'index' or 'statistic' that gives $CO_2$ equivalence. Such a metric can be assessed in radiative forcing terms by the accuracy of the

approximation

$$a_{\text{CO2}}\,\tilde{R}_{\text{CO2}}(p)\,\Delta\tilde{S}_{\text{CO2-eq}}(p) = a_{\text{CH4}}\,\tilde{R}_{\text{CO2}}(p)\,\tilde{\Psi}(p)\,\Delta\tilde{S}_{\text{CH4}}(p) \approx a_{\text{CH4}}\,\tilde{R}_{\text{CH4}}(p)\,\Delta\tilde{S}_{\text{CH4}}(p) \tag{10}$$

If the global temperature response to a change in radiative forcing is linearised using a response function $U(t)$, as is done for example by Myhre et al. (2013), then equivalence in temperature perturbations can be analysed in terms of the approximation

$$\tilde{U}(p)\,a_{\text{CO2}}\,\tilde{R}_{\text{CO2}}(p)\,\Delta\tilde{S}_{\text{CO2-eq}}(p) = \tilde{U}(p)\,a_{\text{CH4}}\,\tilde{R}_{\text{CO2}}(p)\,\tilde{\Psi}(p)\,\Delta\tilde{S}_{\text{CH4}}(p)$$

$$\approx \tilde{U}(p)\,a_{\text{CH4}}\,\tilde{R}_{CH4}(p)\,\Delta\tilde{S}_{\text{CH4}}(p) \tag{11}$$

In both (10) and (11), removing the common factors reduces the comparison to one of considering the accuracy of the approximation

$$\tilde{R}_{\text{CO2}}(p)\,\tilde{\Psi}(p) \approx \tilde{R}_{\text{CH4}}(p) \tag{12}$$

As Wigley (1998) noted 'If $CO_2$-equivalence is based on radiative forcing, and calculated accurately for non-$CO_2$ gases, then the temperature and sea-level implications of the [Kyoto] Protocol may be calculated from the $CO_2$-alone case'.

Because of the commutative and associative properties of such transformations, a transformation of the $CH_4$ source to give an equivalent $CO_2$ source can be described in terms of how well the metric transformation, acting on the $CO_2$ impulse response, reproduces the impulse response for $CH_4$. The application of this relation in the frequency domain (i.e. $p = 2\pi i f$) is described in the appendix.

In these calculations, the response used for $CO_2$ is the multi-model mean from (Joos et al., 2013, Table 5) and the response of $CH_4$ described by a 12.4 year perturbation lifetime (Myhre et al., 2013). In each case, these represent the response to small perturbations about current conditions, reflecting our interest in the use of metrics for trade-offs, reporting and target-setting. The values for $a_{\text{CO2}}$ and $a_{\text{CH4}}$ are also taken from (Myhre et al., 2013) and in the latter case, follow the IPCC convention of including indirect effects. These factors only appear in the relative scaling of the axes in the two parts of Figure 2.

The calculations were developed for methane emissions from active biological sources. For fossil methane, an additional $CO_2$ contribution from the oxidation of $CH_4$, corresponding to a GWP of 1, should be included.

## 3.1 Global Warming Potential

The Global Warming Potential (GWP) with time horizon $H$ defines an equivalence (denoted $\underset{\text{GWP}}{\equiv}$ ) for component Y given by

$$\Delta S_{\text{CO2}}(t) \underset{\text{GWP}}{\equiv} \mathbf{GWP}_H\,\Delta S_{\text{Y}}(t) \tag{13}$$

where

$$\mathbf{GWP}_H = \frac{a_Y}{a_{CO2}} \frac{H^{-1} \int_0^H R_Y(t') \, dt'}{H^{-1} \int_0^H R_{CO2}(t') \, dt'} \quad \text{for gas Y} \tag{14}$$

Although (14) is usually written without the $H^{-1}$ factors, in the form above the numerator and denominator correspond to the airborne fractions of $Y$ and $CO_2$ respectively, averaged over the time horizon $H$, and multiplied by the factor $a_Y/a_{CO2}$ which corresponds to $\mathsf{GWP}_0$, the $H \to 0$ limit of $\mathsf{GWP}_H$. This factor can be called the instantaneous GWP.

$\mathsf{GWP}_{100}$, the GWP with the time horizon $H = 100$ years, has become the standard for greenhouse gas equivalence in international agreements.

For $CH_4$, the equivalence is

$$\Delta S_{CO2}(t) \underset{\text{GWP:100}}{\equiv} \mathsf{GWP}_{100} \Delta S_{CH4}(t) \tag{15}$$

where all use of GWP in what follows will specifically refer to $CH_4$. Relation (15) corresponds to using

$$\tilde{R}_{CH4}(p)/\tilde{R}_{CO2}(p) \approx \tilde{\Psi}_{GWP}(p) = \mathsf{GWP}_{100}/\mathsf{GWP}_0 \tag{16}$$

which is plotted as the horizontal line (long dashes) in Figure 1.

However, this definition of equivalence has long been known to be poor (e.g. Wigley, 1998; Reilly et al., 1999), especially for emission profiles approaching stabilisation of concentrations.

For $H > 100$ the approximation

$$\tilde{R}_{CH4}(p)/\tilde{R}_{CO2}(p) \approx \mathsf{GWP}_{H=1/p}/\mathsf{GWP}_0 \tag{17}$$

is quite close (Enting, 2018). Thus in the context of emissions $\Delta S_{CH4}$ growing with $e$-folding time, $H$, $\mathsf{GWP}_H$ gives approximate FEI equivalence. Specifically $\mathsf{GWP}_{100}$ gives approximate equivalence for 1% per annum growth rate and, as shown in Figure 1, about a 30% underestimate for the 2% per annum growth rate that approximately characterises 20th century changes.

## 3.2 Derivative

Several studies (Smith et al., 2012; Lauder et al., 2013) suggested that for short-lived gases such as $CH_4$, changes in emissions in the short-lived gases should be related to one-off $CO_2$ emissions. This suggests a metric of the form:

$$\Delta S_{CO2}(t) \underset{\text{DERIV}}{\equiv} 100 \, \mathsf{GWP}_{100} \frac{d}{dt} \Delta S_{CH4}(t) \tag{18}$$

or (as a Laplace transform):

$$\tilde{R}_{CH4}(p)/\tilde{R}_{CO2}(p) \approx \tilde{\Psi}_{Deriv}(p) = 100 \, p \, \mathsf{GWP}_{100}/\mathsf{GWP}_0 \tag{19}$$

which is plotted as the straight line through the origin (chain curve) in Figure 1.

Subsequently, the search for an improved metric, termed GWP*, has been the subject of extensive studies undertaken by Allen and co-workers: (Allen et al., 2016, 2018; Jenkins et al., 2018; Cain et al., 2019; Collins et al., 2020; Lynch et al., 2020). These studies have included cases defined by linear combinations of the derivative metric and GWP. Such cases are not shown in the transform domain illustrated in Figure 1, but correspond to linear functions of $p$ that do not pass through the origin.

### 3.3 Difference

A recent proposal for an improved GWP* (Cain et al., 2019) defined the equivalence:

$$\Delta S_{\text{CO2}}(t) \underset{\text{DIFF}}{\equiv} \text{GWP}_{100}\left[4\Delta S_{\text{CH4}}(t) - 3.75\Delta S_{\text{CH4}}(t-20)\right] \tag{20}$$

The Laplace transform is derived using the generic result that a time-shift by $T$ corresponds to multiplying the Laplace transform by $\exp(-pT)$, giving:

$$\tilde{R}_{\text{CH4}}(p)/\tilde{R}_{\text{CO2}}(p) \approx \tilde{\Psi}_{\text{Diff}}(p) = \text{GWP}_{100}/\text{GWP}_0 \times [4 - 3.75\exp(-20p)] \tag{21}$$

This is shown in Figure 1 using the short dashes.

### 3.4 Reduced model

When, as is done here, the response functions are expressed as sums of exponentially decaying functions of time, the Laplace transforms become sums of partial fractions of the form $\alpha/(p+\beta)$ so that the combination is a ratio of polynomials in $p$. Thus the FEI ratio will also be a ratio of polynomials which can in turn be re-expressed as a sum of partial fractions. Formally this gives an exact form for the FEI relation but one which would have perhaps 6 to 10 parameters and be too complicated for practical use. Studies in a number of fields such as electronic engineering (e.g. Feldman and Freund, 1995) have noted that such expressions can often be usefully approximated by lower-order expressions. For emission equivalence, it is only practical to use very low-order approximations for such a reduced model.

As shown in Figure 1, a close fit to FEI can be obtained with the reduced model (RM) given by

$$\tilde{R}_{\text{CH4}}(p)/\tilde{R}_{\text{CO2}}(p) \approx \tilde{\Psi}_{\text{RM}}(p) = \frac{p}{p+b} \tag{22}$$

with $b = 0.035$ which is plotted as the dotted curve in Figure 1.

This gives an equivalence:

$$\frac{a_{\text{CH4}}}{a_{\text{CO2}}}\frac{p}{p+b}\Delta\tilde{S}_{\text{CH4}}(p) \underset{\text{RM}}{\equiv} \Delta\tilde{S}_{\text{CO2}}(p) \tag{23}$$

In the time domain, (23) becomes:

$$\frac{a_{\text{CH4}}}{a_{\text{CO2}}}\int_0^t \exp(-b(t-t'))\Delta\dot{S}_{\text{CH4}}(t')\,dt' + \frac{a_{\text{CH4}}}{a_{\text{CO2}}}\Delta S_{\text{CH4}}(t=0)\exp(-bt) \underset{\text{RM}}{\equiv} \Delta S_{\text{CO2}}(t) \tag{24}$$

where $\Delta \dot{S}_{CH4}$ denotes the rate of change in the perturbation to $CH_4$ emissions.

This expresses the $CO_2$-equivalent of $CH_4$ as a weighted average of the $CH_4$ emission growth rate. Consequently, the metric retains the property that constant emissions of $CH_4$ are treated as equivalent to zero $CO_2$ emissions as in 'derivative' metrics (Smith et al., 2012; Lauder et al., 2013). The parameter $b$ can be chosen to match other metrics. The value $b = 0.035$ is chosen so that for emissions with 1% per annum growth rate the RM metric closely matches the 100-year GWP.

For specific calculations it may be more appropriate to represent this metric as

$$\frac{a_{CH4}}{a_{CO2}} \left[ \Delta S_{CH4}(t) - b \int_0^t \exp(-b(t-t')) \Delta S_{CH4}(t')\, dt' \right] \underset{RM}{\equiv} \Delta S_{CO2}(t) \tag{25}$$

Relation (25) is derived from (24) using integration by parts (or equivalently by putting $p/(p+b) = 1 - b/(p+b)$). It has the advantage that it is expressed in terms of emissions rather than their rates of change.

Equation 25 defines the reduced model equivalence as a difference between present emissions and a weighted average of past emissions. When considered in terms of frequency $f$ (by setting $p = 2\pi f \times \sqrt{-1}$) this avoids the frequency aliasing that occurs with the 'difference' metric for periods of 20 years or integer fractions thereof (see Figure 3 in the appendix).

The equivalence relation (23) can also be re-written as

$$p\Delta \tilde{S}_{CH4}(p) \underset{RM}{\equiv} \frac{a_{CO2}}{a_{CH4}}(p+b)\Delta \tilde{S}_{CO2}(p) \tag{26}$$

This defines an equivalence between the rate of change of $CH_4$ emissions and a combination of rate of change of $CO_2$ emissions (as in GWP) and current $CO_2$ emissions (as in the derivative-based equivalences suggested by Smith et al. (2012) and Lauder et al. (2013)).

## 4 Comparisons in the time domain

Many previous studies of metrics have concentrated on global-scale calculations over the long term. As discussed above, this has led to the development of metrics based on rates of change. However, as discussed in Section 5 below, for emissions trading on shorter timescales, political acceptance is likely to favour metrics that also have equivalent influences in the short term. The short-term behaviour can be analysed by taking a notional $CH_4$ emission profile and calculating the resulting $CH_4$ concentrations. This is then compared to the $CO_2$ concentrations that result from the notionally equivalent $CO_2$ emissions.

Figure 2(a) shows a $CH_4$ source perturbation with a rapid increase from zero to a fixed emission rate, and the $CO_2$-equivalent emissions as determined by the various equivalence metrics. Figure 2(b) shows the $CH_4$ concentration resulting from the methane emissions and the $CO_2$ concentration resulting from the various $CO_2$-equivalent emissions. In Figures 2 (a) and (b), the relative scaling of the axes is given by $a_{CH4}/a_{CO2}$ so that forcing can be compared directly. In this scaling, the direct effect of $CH_4$ has been scaled to include indirect effects, from tropospheric ozone and stratospheric water vapour, using values taken from Myhre et al. (2013). Note that the indirect effects are not included in the corresponding graphs given by Enting and Clisby (2020).

The results in Figure 2 (b) clearly show the failings of the 100-year GWP for defining emission equivalence for constant sources. The forcing from GWP-equivalent $CO_2$ (long dashes) initially lags well behind the actual forcing from $CH_4$ but in the long term it continues to increase indefinitely long after the forcing from on-going $CH_4$ emissions has stabilised. Compared to this behaviour, the 'derivative' metric based on rates of change of $CH_4$ emissions is a great improvement (chain curve).

However, the $CO_2$-equivalent forcing initially exceeds the actual forcing from $CH_4$ and in the long-term drops below the $CH_4$ forcing. The difference metric from Cain et al. (2019) (short dashes) provides a $CO_2$-equivalent forcing that follows the actual $CH_4$ forcing more closely with only a slight shortfall in the longer term. After $t = 150$ the forcing from equivalence defined by the Cain et al. (2019) metric (short dashes) starts to increase, due to the contribution that corresponds to 0.25 times GWP when $S_{CH4}(t) \approx S_{CH4}(t-20)$.

Figure 2 (b) shows that the $CO_2$-equivalence derived from the reduced model (dotted curve) follows the actual $CH_4$ forcing particularly closely, as would be expected given the close agreement when the relations are expressed as Laplace transforms as shown in Figure 1.

The nature of the FEI relation precludes close matches in forcing from instantaneous relations between $CH_4$ and $CO_2$ emissions. The 'difference' and 'reduced model' metrics relate $CO_2$ equivalents to the past history of $CH_4$ emissions. For a specific case, Lauder et al. (2013) suggested an approximate equivalence to step changes in methane emissions balanced by an ongoing future $CO_2$ uptake from growing trees.

## 5 Practical issues for implementation

The aim of our analysis has been to provide a better understanding GWP vs GWP* and similar metrics. Any comprehensive analysis of what might be politically feasible needs to be done by others with greater expertise in such areas. However, there are various aspects of our analysis that bear on the practical applicability and political acceptability of various metrics and the trade-offs that need to be balanced in political choices.

Past studies cited above suggest that an equivalence metric should capture the context of emissions at the time. The analysis by Enting (2018) (see also equation 17 above) notes that $GWP_H$ is close to FEI equivalence for growth in emissions with an $e$-folding time of $H$. Thus a 100-year GWP was a plausible approximation at the time that it was introduced. For very large $H$, the GWP of short-lived gases goes to zero as $1/H$ suggesting that a derivative of growth rates should define the metric for such long timescales. In contrast, for short-term trading and target setting, a metric that captures the short term context is desirable in order to avoid distortions that would hinder political acceptability.

An important goal of defining emissions equivalence is to allow for emissions of different greenhouse gases to be substituted for each other, so that a given target expressed in terms of radiative forcing (or equivalently in terms of $CO_2$ concentration equivalence) can be achieved for the least economic cost. If, as is the case for $GWP_{100}$, the metric over-estimates the extent to which $CO_2$-equivalent emission reductions contribute to radiative forcing, then methane reductions based on such equivalence will fall short of the $CO_2$ concentration-equivalent target. Conversely, for short timescales where $GWP_{100}$ under-estimates to

forcing reduction of $CO_2$-equivalent methane reductions, short-term targets based on such equivalence will over-estimate the extent of requisite methane emission reductions as in the example given by Wigley (1998).

In considering how our analysis feeds into such considerations, we note:

- the metric should capture both the long-term context needed for stabilisation and the more immediate context in which both trading and international agreements are conducted;

- if the metric for emissions equivalence is too complex, as it is for FEI, then it may be difficult or impossible for an effective trading scheme to be implemented;

- the metric needs to be 'backward looking' and avoid giving present credit or debit on the basis of promises of future targets;

- however the backwards view should not extend too far as the relevant actors can change over time, even in the cases of nations or multi-national groups, such as the EU which has in the past set collective targets;

- metrics defined in terms of derivatives need to be supplemented with a specification of how this is determined in practice
e.g. as difference by Cain et al. or the transformation from equation (24) in terms of rates of change of sources to equation (25) in terms of actual sources for the Reduced Model metric.

    Finally, we note that our analysis is illustrative, using specific numbers primarily from the 5th IPCC assessment. The forthcoming 6th IPCC assessment may well make minor changes to specific numbers such as the effective lifetime and the $CO_2$ response fucntion as well as such things as the inclusion of feedbacks, forcing efficiencies and indirect effects (cf Myhre et al.,
2013).

## 6    Concluding summary

Our analysis has used the concept of FEI-equivalence to analyse various definitions of greenhouse gas emission equivalence in terms of how closely equivalent emissions at a time $t$ lead to equal radiative forcing at future times. The approach is applied to the consideration of $CH_4$ emissions in terms of various definitions of their $CO_2$-equivalent emissions. In the special case of
exponentially growing emissions, FEI-equivalence can be achieved when the emissions are scaled by the instantaneous (0 time horizon) GWP, multiplied by the ratio of the asymptotic airborne fractions. This ratio depends on the $e$-folding growth rate. Various emission metrics can be compared in terms of how well they match this ratio at the range of relevant timescales. This analysis is equivalent to considering Laplace transforms of the impulse response functions of the respective gases.

    GWP treats this ratio as a constant for all timescales, effectively defining $GWP_H$ as the instantaneous GWP multiplied by the
ratio of average airborne fractions over the time horizon, $H$. For $CH_4$, referenced to $CO_2$, this means that GWP over-estimates the $CH_4$ contribution for growth rates less than $1/H$ and under-estimates the $CH_4$ contribution to radiative forcing at faster growth rates.

Metrics relating $CO_2$-equivalence to rates of change of $CH_4$ emissions, or emissions of other short-lived gases, are treating the ratio of airborne fractions as proportional to the $e$-folding rate. This can provide a good representation of long-term behaviour relevant for stabilisation, but over-estimates the role of $CH_4$ on the shorter timescales relevant for emission trading

A range of metrics that better match FEI over a wide range of timescales from decades to millennia can be constructed. These include the metric proposed by Cain et al. (2019) which uses the change in $CH_4$ emissions over a 20 year interval, and a reduced model approximation to FEI-equivalence. In each of these cases the better match is achieved at the expense of comparisons involving longer time periods.

The political acceptability of metrics other than the GWP will involve various trade-offs between accuracy and practicality. The type of analyses presented here can help analyse such trade-offs without reference to specific scenarios of changes in greenhouse gas emissions.

*Code availability.* The R code used to perform the calculations and generate the figures is archived in FigShare with doi 10.6084/m9.figshare.13667657.

**Appendix: Frequency domain analysis**

The Laplace transform provides a natural formalism for analysing causal initial value systems. However Fourier transforms and Fourier analyses have wide familiarity and can be used to describe our results.

For a periodic variation with exponentially increasing amplitude, equation (5) generalises to

$$\int_{-\infty}^{t} \exp(\alpha t' + i\omega t') R(t-t') \, dt' = \exp(\alpha t + i\omega t) \int_{0}^{\infty} R(t') \exp(-\alpha t' - i\omega t') \, dt' \tag{1}$$

For $R_{CO2}$, this relation requires $\alpha > 0$ in order to have the lower limit of the left-hand integral and the upper limit of the right hand integral defined. The $\alpha \to 0$ limit shows the relation between the Laplace transform and the Fourier transform, which, for functions with $R(t) = 0$ for $t < 0$, is given by the integral on the right.

Section 3 noted that metric transformations defined by

$$a_{CO2} \, \tilde{S}_{CO2\text{-eq}}(p) = a_{CH4} \, \tilde{\Psi}(p) \, \tilde{S}_{CH4}(p) \tag{2}$$

can be assessed in radiative forcing terms by the accuracy of the approximation

$$a_{CO2} \, \tilde{R}_{CO2}(p) \, \tilde{S}_{CO2\text{-eq}}(p) = a_{CH4} \, \tilde{R}_{CO2}(p) \, \tilde{\Psi}(p) \, \tilde{S}_{CH4}(p) \approx a_{CH4} \, \tilde{R}_{CH4}(p) \, \tilde{S}_{CH4}(p) \tag{3}$$

which reduces to comparing

$$\tilde{R}_{CO2}(p) \, \tilde{\Psi}(p) \approx \tilde{R}_{CH4}(p) \tag{4}$$

where for FEI equivalence, the approximation becomes exact equality.

A frequency domain interpretation can be obtained by putting $p = 2\pi i f$. In these terms, the metric transformation is acting like a frequency equaliser in an audio system.

The phases of the complex numbers in the relations above capture the phase shifts for the various frequencies. For the present we show only the resulting amplitudes, given by the moduli, $|z|$, of the complex value, and ignore the phase (noting that the modulus of a product is the product of the moduli).

Figure 3 sets $p = 2i\pi f$ to evaluate the various cases considered in the paper, as functions of frequency $f$ in cycles per year. It shows

- $|\tilde{R}_{CH4}(p)|$, the 'target' for FEI equivalence; the zero frequency value is the perturbation lifetime;

- $|\tilde{R}_{CO2}(p)\tilde{\Psi}_{GWP}(p)|$, i.e. a multiple of the $CO_2$ response, growing indefinitely as frequency goes to zero;

- $|\tilde{R}_{CO2}(p)\tilde{\Psi}_{Deriv}(p)|$ which gives a better approximation over a wider range of frequencies;

- $|\tilde{R}_{CO2}(p)\tilde{\Psi}_{Diff}(p)|$ which gives a further improvement, but a notable discrepancy for cycles whose period is near the 20-year interval used in the difference calculation;

- $|\tilde{R}_{CO2}(p)\tilde{\Psi}_{RM}(p)|$ which gives a still closer fit over the range of frequencies shown.

**Appendix: Notation**

Laplace transforms are denoted by the tilde notation with $\tilde{R}(p)$ as the Laplace transform of $R(t)$.

Equivalence relations are denoted by $\equiv$ with particular cases identified, e.g. $\underset{\text{GWP}}{\equiv}$ .

$a_X$  Radiative forcing per unit mass of constituent $X$.

$b$  e-folding rate in reduced model equivalence relation.

$F_X(t)$  Radiative forcing of constituent $X$.

**GWP**, **GWP**$_H$  Global warming potential for $CH_4$ (unless otherwise specified), for time horizon $H$.

$H$  Time horizon for GWP.

$M_X(t)$  Atmospheric mass of constituent $X$. Perturbation is $\Delta M_X(t)$.

$p$  Argument of Laplace transform. Equivalent to $e$-folding rate when comparing exponentially growing emissions.

$R_X(t)$  Atmospheric response function for constituent $X$.

$S_X(t)$  Anthropogenic emission of constituent $X$. Perturbation is $\Delta S_X(t)$.

$t$  Time

X,Y  Labels for constituent. Specific cases $CO_2$, $CH_4$.

$\alpha$  $e$-folding rate of exponentially growing emissions.

$\delta(t)$ Delta 'function'. Instantaneous unit pulse. The notional derivative of unit step function.

$\tilde{\Psi}(p)$ Laplace transform of generic integro-differential operator that defines a metric transformation. Specific instances are $\tilde{\Psi}_{\text{FEI}}$, $\tilde{\Psi}_{\text{GWP}}$, $\tilde{\Psi}_{\text{Deriv}}$, $\tilde{\Psi}_{\text{Diff}}$ and $\tilde{\Psi}_{\text{RM}}$.

*Author contributions.* Both authors worked on the mathematical analysis, the computer code and the writing and checking of the manuscript.

*Competing interests.* The authors have no competing interests.

*Acknowledgements.* The authors gratefully acknowledge the contribution of Alan Lauder in bringing the issue of $CH_4$ vs $CO_2$ comparisons to our attention. We also wish to thank Annette Cowie for valuable comments on the manuscript. We also acknowledge helpful review comments from William Collins and the anonymous referee.

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

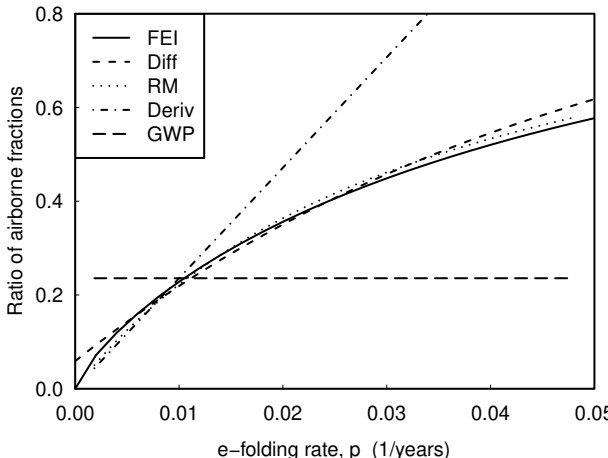

**Figure 1.** Ratio of airborne fractions for $CH_4$ relative to $CO_2$ as defined or assumed for various metrics. The solid curve shows the FEI which acts as a reference. The GWP line treats this ratio as independent of timescale (eqn 16); the chain line for the 'Deriv' case treats the timescale dependence as proportional to the inverse timescale (eqn 19); the shorter dashes of the Diff curve (eqn 21) more closely approximate FEI. The dotted line, 'RM', is an empirical 'reduced model' approximation (eqn 23) to FEI. These curves can also be interpreted as the Laplace transforms of the operations that define the equivalence in the time domain.

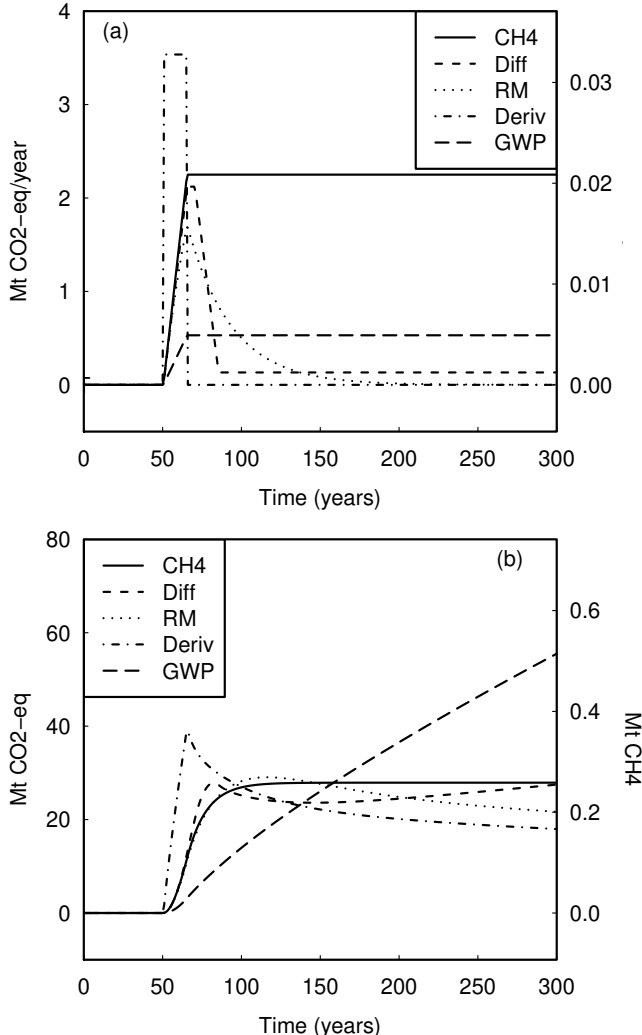

**Figure 2.** (a) A CH$_4$ source representing an increase, over 15 years, from zero to a constant (solid line) and the CO$_2$-equivalent sources as defined by the various metrics described in Section 3. The relative scaling of the CH$_4$ and CO$_2$ axes is $a_{CH4}/a_{CO2}$. (b) CH$_4$ concentrations from source shown in part (a) (solid line) and the CO$_2$ concentrations resulting from the CO$_2$ concentrations resulting from the equivalent CO$_2$ sources, as shown in part (a). The relative scaling of the axes is $a_{CH4}/a_{CO2}$ so that the radiative forcing can be compared directly.

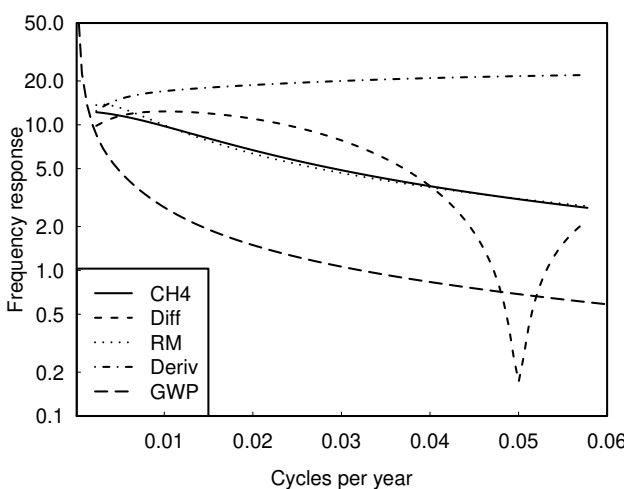

**Figure 3.** Frequency response for the various cases of $|\tilde{R}_{CO2}(p)\,\tilde{\Psi}(p)|$ discussed above, compared to the actual frequency response, $|R_{CH4}(p)|$ to periodic CH$_4$ emissions (solid line), using $p = 2\pi i f$.