# Peer review of "Technical note: On comparing greenhouse gas emission metrics"

_Atmospheric Chemistry and Physics, 2020_

## Referee Comment (RC1) · Anonymous Referee #1 · 17 Nov 2020

General remarks:

1) I could not find the definitions of the manuscript type "technical note" on the ESD website (https://www.earth-system-dynamics.net/about/manuscript_types.html), so I cannot comment on the style of the submitted manuscript.

2) For the mathematically versed, Enting and Clisby provide some gourmet material, I am sure. I am myself not well versed in Laplace transforms, so I cannot comment on those technical aspects of the manuscript and would hope that the reviewer pool contains a knowledgeable person in that regard.

3) Aside from the mathematical beauty of the discussed equivalence metrics: IMHO, this branch of equivalence metrics has gone down a politically completely impractical

rabbit hole. Providing credits or debits to a country or actor under the implicit assumption that that country or actor will keep its emissions constant for all times, is as far away from political realities and practicalities as it can be. The reason is that the metric value of a few hundred or a few thousand (the value of lowering the level of CH4 emissions compared to a one-off emission of CO2) would make countries focus 100% on methane reductions (which would negate any need to do anything else, while methane emissions can be lowered) and as soon as methane emissions cannot be lowered, but increase, e.g. as agricultural production increases etc, the country would face such a heavy "penalty" that it cannot possibly offset that methane increase within a target period (e.g. under the Paris Agreement). Thus, the country would drop out of the international regime. In summary, these emission-rate focussed equivalence metrics like GWP*, while solving a scientific, mathematical problem nicely, would make any international effort of reducing a basket of GHGs politically impractical.

Detailed comments:

Line 64: Provide the reference for the assumed CH4 methane lifetime.

Line 66: Clarify whether the authors suggest adding "one additional CO2 contribution", i.e. increasing the metric value of CH4 by 1 or another value.

Line 73: The linearisation of CO2 forcing – by implying F = alpha * R (the airborne mass) is not necessarily in line with recent line-by-line codes (Etminan et al., 2016). While a valid linearisation for small deviations of emissions, please bring that to the reader's attention.

Line 87: Apologies, that I do not get this. More explanation of this line would be useful.

Line 123: Why should the parameter b be dependent of an annual growth rate – and what would be the advantage of it matching 100-year GWPs in an exponential growth scenario? Explain. . .

Line 137: it is unclear to me what the authors mean with "When metrics are used

for emissions trading, the behaviour at shorter timescales becomes important.". The timescale of interest for a metric is ideally roughly representative with the objective function that policy-makers would like to pursue: like minimising climate change over the next 100 years. Or to limit peak warming in 40 years from now etc. Emission trading only operationalises the emphasis that you would place on the mitigation of one GHG over another. Emission trading itself does not favour "shorter timescales" in terms of the metrics. The authors should explain / clarify what they mean.

Line 145: "in this type of context" is imprecise. The 100-year GWP does, as Figure 3 sort of suggests, pretty much exactly what it promises to do: Creating emission equivalence in terms of cumulative radiative forcing over the 100-year time horizon.

Line 151: The sentence "The increase after several centuries reflects. . . " is unclear. What does it refer to? Please clarify.

Line 174: It is unclear to me why the authors write that for GWP-H, with H being the time horizon, say H = 100yrs, "GWP . . . under-estimates the CH4 contribution from shorter timescales". Well, the GWP-100 is defined that the integral of radiative forcing should be the same from a unit CO2 and a unit CH4 emissions over a 100-year time horizon. So, by design, GWP will then under-estimate the CH4 contribution from timescales that are shorter than the cross-over point (0>x>100), and afterwards over-estimate the CH4 contribution. If CH4 "contribution" refers to the radiative forcing. Please either clarify, correct, or both ..

Line 159: The authors write "The goal of defining emission equivalence is to allow for emissions of different . . . so that a given radiative forcing target can be achieved for the least economic cost". Well, metrics have various explicit or implicit objective functions, but "radiative forcing targets" is not usually one. It is either cumulative radiative forcings (GWPs), which approximately the integrated warming over a certain period of time, or it is warming at a particular point in time (GTPs) etc.. Please clarify / rephrase.

Line 156: It is unclear what the authors want to say with the sentence "For a specific case, Lauder et al. (2013) suggested an approximate equivalence to changes in methane emissions balanced by an ongoing future CO2 uptake from growing trees". It is unclear whether the authors support that conclusion. In the light of the above discussion, with zero-CO2 emissions being 'equivalent' with non-changing CH4 emission rates (after a CH4 concentrations reach their new equilibrium), whether the Lauder et al. conclusion (as presented here) still holds. Please clarify.

Line 165: In the conclusion, I'd appreciate a bit of discussion from the authors on the general point I raise above, i.e. the practicality of rate-based emission metrics vs GWP-100 in a real-world context.

Figure 3: Maybe I misunderstand the scaling of the y-axes, but the dashed GWP line should not cross the solid CH4 line in year 100, but earlier. The GWP-equivalence is given if the cumulative radiative forcing over a 100-year time horizon is equal. Thus, the crossing of the dashed line should be such that the integral underneath the solid and the dashed (GWP) line is identical from year 50 to year 150. Please clarify or explain why I am wrong in thinking that.

---

## Referee Comment (RC2) · William Collins (Referee) · 23 Nov 2020

This technical note provides an interesting theoretical analysis of CO2 forcing equivalence. Even though the note is designed to be mainly technical it poses many issues that require more discussion here.

The analysis provides a nice comparison of different metrics using a Laplace framework. It seems that the forcing equivalent CO2 emissions can be expressed as a Reduced Model in equation (20). It should be explained if this is a simpler methodology than inverting the CO2 response function as in Wigley 1998.

It is not clear from this paper how this could be applied in any policy context. Typically emission metrics are presented and used as a single number (or two numbers, short

and long, in Levasseur et al. papers). The authors should explain in what context a continuous function Delta_S_CO2(t) could be useful as a metric.

It appears (like Cain et al. 2019) that the metric depends on the past emission history. This means that the larger the past emissions the lower the metric. This could be controversial politically and at least some short discussion is warranted on how/why past behaviour influences the future, and what the implications might be for policy.

Given the similarities of equations 15 and 20, it would be useful to compare them more fully. Do the two terms on the left hand side of equation 20 correspond to the two terms on the right hand side of equation 15? Can the 4 and 3.75 coefficients in equation 15 be related to the b coefficient in equation 20?

The conclusions would be better as flowing text, rather than the series of bullet-like points.

Line 16: I'm not sure "so-called" is a necessary qualifier for greenhouse gases.

Line 21: a_x needs to be defined.

Sections 3.2 and 3.3 need to refer to figure 1 and it needs to be clearer which lines in the figure are being referred to in these sections.

Section 3.4: The key parameter here is "b" so there needs to be more explanation of what this might relate to physically. The text explains a derivation for a 1%/yr growth rate. Would b be completely different for a different emission profile (e.g. figure 2)?

Line 129: This sentence about frequency aliasing is too cryptic as written here. This either needs to be expanded or removed.

Line 136: Why does the behaviour at shorter timescales become important when used for emission trading?

Figures 2 and 3: Since these need to be viewed together I suggest combing into two panels (a) and (b) of figure 2.

Line 164: Some explanation is needed why "least cost" overshoots the radiative forcing target for GWP100.

Conclusions: I found this bullet style very difficult to read or to pull out the key points. I suggest rewriting completely and focussing on the key points as to what has been concluded, and what should a reader take away.

Line 174: It might be better to write as "faster growth rate", since "shorter timescales" might be confused with GWP20 etc., and line 177. Also I'm not sure this is very policy-relevant as no plausible future emission scenario has exponentially growing methane emissions.

177: I think "shorter timescales" here means something different to line 174. It is not obvious why the ratio of airborne fractions is a good representation of long-term behaviour, but not for emissions trading.

Supplementary material The text and code need to be separated here. It was very difficult to follow the text when it was so broken up by code.

---

## Author Comment (AC1) · 5 Jan 2021

We are posting detailed responses to each of the reviewers. However their comments prompt us to make two large scale changes:

1. A specific section on how aspects of our analysis bear on the practicalities and political feasibility of various metrics. In part this will draw on existing text.

2. An appendix giving aspects of a corresponding analysis in the frequency domain. Although the mathematics is a little more complex, expressing the results as frequency analysis may be more familiar to many people than is the Laplace transform. In part this will draw on text from the supplementary information.

In addition, the concluding section has been restructured as suggested by reviewer 2.

[Figure]

These will be posted separately.

In addition we note that the relative axis scaling of Figures 2 and 3 (which will be combined into 2a and 2b, following the suggestion of reviewer 2) reflects GWP defined without the indirect effects. We intend to rescale these axes to reflect GWP calculated with indirect effects — the convention now used by the IPCC. This changes only the axis labelling and does not affect the relative position of any of the lines.

---

## Author Comment (AC2) · 5 Jan 2021

**Conclusions, restructured as suggested by reviewer 2**

Our analysis has used the concept of FEI-equivalence to analyse various definitions of greenhouse gas emission equivalence in terms of how closely equivalent emissions at a time $t$ lead to equal radiative forcing at future times. The approach is applied to the consideration of $CH_4$ emissions in terms of various definitions of their $CO_2$-equivalent emissions. In the special case of exponentially growing emissions, FEI-equivalence can be achieved when the emissions are scaled by the instantaneous (0 time horizon) GWP, multiplied by the ratio of the asymptotic airborne fractions. This ratio depends on the $e$-folding growth rate. Various emission metrics can be compared in terms of how

well they match this ratio at the range of relevant timescales. This analysis is equivalent to considering Laplace transforms of the impulse response functions of the respective gases.

GWP treats this ratio as a constant for all timescales, effectively defining $\text{GWP}_H$ as the instantaneous GWP multiplied by the ratio of average airborne fractions over the time horizon, $H$. For $CH_4$, referenced to $CO_2$, this means that GWP over-estimates the $CH_4$ contribution for growth rates less than $1/H$ and under-estimates the $CH_4$ contribution from shorter timescales.

Metrics relating $CO_2$-equivalence to rates of change of $CH_4$ emissions, or emissions of other short-lived gases, are treating the ratio of airborne fractions as proportional to the $e$-folding rate. This can provide a good representation of long-term behaviour relevant for stabilisation, but over-estimates the role of $CH_4$ on the shorter timescales relevant for emission trading

A range of metrics that better match FEI over a wide range of timescales from decades to millennia can be constructed. These include the metric proposed by Cain et al. (2019) which compares $CH_4$ emissions over a 20 year interval, and a reduced model approximation to FEI-equivalence, achieved at the expense of comparisons involving longer time periods.

The political acceptability of metrics other than the GWP will involve various trade-offs between accuracy and practicality. The type of analyses presented here, can help analyse such trade-offs without reference to specific cases of changes in greenhouse gas emissions.
* * *

---

## Author Comment (AC3) · 5 Jan 2021

**Author comment: Proposed additional section - Practicalities for implementation**

The aim of our analysis has been to provide a better understanding GWP vs GWP* and similar metrics. Any comprehensive analysis of what might be politically feasible needs to be done by others with greater expertise in such areas.

However, there are various aspects of our analysis that bear on the practical applicability and political acceptability of various metrics and the trade-offs that need to be balanced in political choices.

Past studies suggest that an equivalence metric should capture the context of emissions at the time. The analysis by Enting (2018) (see equation 12 above) notes that

$GWP_H$ is close to FEI equivalence for a growth in emissions with an $e$-folding time of $H$. Thus a 100-year GWP was a plausible approximation at the time that it was introduced. For very large $H$, the GWP of short-lived gases goes to zero as $1/H$ suggesting that a derivative of growth rates should define the metric for such long timescales. In contrast, for short-term trading and target setting, a metric that captures the short term context is desirable in order to avoid distortions that would hinder political acceptability.

An important goal of defining emissions equivalence is to allow for emissions of different greenhouse gases to be substituted for each other, so that a given target expressed in terms of radiative forcing (or equivalently in terms of $CO_2$ concentration equivalence) target can be achieved for the least economic cost. If, as is the case for GWP100, the metric over-estimates the extent to which $CO_2$-equivalent emission reductions contribute to radiative forcing, then methane reductions based on such equivalence will fall short of the $CO_2$ concentration-equivalent target. Conversely, for short timescales where GWP100 under-estimates to forcing reduction of $CO_2$-equivalent methane reductions, short-term targets based on such equivalence will over-estimate the extent of requisite methane emission reductions as in the example given by Wigley (1998).

In considering how our analysis feeds into such considerations, we note:

- the metric should capture both the long-term context needed for stabilisation and the more immediate context in which both trading and international agreements are conducted;

- if the metric for emissions equivalence is too complex, as it is for FEI, then it may be difficult or impossible for an effective trading scheme to be implemented;

- the metric needs to be 'backward looking' and avoid giving present credit or debit on the basis of promises of future;

- however the backwards view should not extend too far as the relevant actors can change over time, even in the cases of nations or even multi-national groups,

such as the EU which has in the past set collective targets;

- metrics defined in terms of derivatives need to be supplemented with a specification of how this is determined in practice e.g. as difference by Cain et al. or the transformation from equation (19) in terms of rates of change of sources to equation (20) in terms of actual sources for the Reduced Model metric.

Finally, we note that our analysis is illustrative, using specific numbers primarily from the 5th IPCC assessment. The forthcoming 6th IPCC assessment may well make minor changes to specific numbers, effective lifetime and $CO_2$ response as well as such things as the inclusion of feedbacks, forcing efficacies and indirect effects (Myrhe et al, 2013. IPCC AR5 WG1 Ch 8).

---

## Author Comment (AC4) · 5 Jan 2021

Author comment: Response to referee 2

We are not proposing a continuous metric. $\Delta S_{CO2}(t)$ is not a metric. It is the result, for a particular time $t$, of applying a metric. The metric is the process of going from $\Delta S_{CH4}(.)$ to the CO2-equivalent. In general mathematical terms this would be a functional. Restricting such functionals to time-invariant linear operators whose Laplace transforms are rational functions restricts consideration to metrics defined as linear integro-differential operators. The full inversion of the Wigley FEI relation can be expressed in this way (most easily by using Laplace transforms) if the $CH_4$ and $CO_2$ responses are sums of exponentials. However, our analysis suggests that useful ap-

proximations can be obtained using much simpler expressions.

The various metric processes that we consider for generating $\Delta S_{CO2}(t)$, each applicable at any single time $t$, are

- multiply $\Delta S_{CH4}(t)$ by a constant (ie, GWP approach)

- multiply $\frac{d}{dt}\Delta S_{CH4}(t)$ by a constant — in practice this would require a specification of how the derivative is defined

- combine current $\Delta S_{CH4}(t)$, with the 20-year difference $4\Delta S_{CH4}(t) - 3.75\Delta S_{CH4}(t-20)$

- take current $\Delta S_{CH4}(t)$ offset by weighted integral over past emission perturbations.

Line by line comments

**Line 16** the use of 'so-called' captures the fact that actual greenhouses don't work by changing radiation balance. Our bending over backwards for correctness reflects the politicisation of climate science, particularly in Australia and the USA.
**Proposed change:** *Leave decision to editor.*

**Line 21** noted
**Proposed change:** *where $a_X$ is the radiative efficacy in mass units: the amount of change in radiative forcing per unit mass increase for constituent $X$ in the atmosphere.*

**Sec 3.2, 3.3** agreed
**Proposed change:** *'insert as shown by the *** line,' after lines 82, 93, 104, 114. Similar change also made in Section 4.*

**Sec 3.4** We regard the parameter $b$ as being an empirical fit that has no specific physical meaning. The reduced model is fitting the ratio of two response functions whose parameters are themselves empirical fits whose parameters have only distant connection to the underlying processes involved. However, the important point is that $b$ is independent of the growth rate used in the example.

**Proposed change:** *Propose inserting, after line 65:*

A general linear, time-invariant equivalence relation defined by

$$a_{CO2}\Delta\tilde{S}_{CO2-eq}(p) = a_{CH4}\tilde{\Psi}(p)\Delta\tilde{S}_{CH4}(p) \qquad AC2.1$$

can be assessed in radiative forcing terms by the accuracy of the approximation

$$a_{CO2}\tilde{R}_{CO2}(p)\Delta\tilde{S}_{CO2-eq}(p) = a_{CH4}\tilde{R}_{CO2}(p)\tilde{\Psi}(p)\Delta\tilde{S}_{CH4}(p) \approx a_{CH4}\tilde{R}_{CH4}(p)\Delta\tilde{S}_{CH4}(p)$$
$$AC2.2$$

If the global temperature response is linearised using a response function $U(t)$, as in done for example in AR5-WG1-Ch8, then equivalence in temperature perturbations can be analysed in terms of the approximation

$$\tilde{U}(p)a_{CO2}\tilde{R}_{CO2}(p)\Delta\tilde{S}_{CO2-eq}(p) = \tilde{U}(p)a_{CH4}\tilde{R}_{CO2}(p)\tilde{\Psi}(p)\Delta\tilde{S}_{CH4}(p)$$

$$\approx \tilde{U}(p)a_{CH4}\tilde{R}_{CH4}(p)\Delta\tilde{S}_{CH4}(p) \qquad AC2.3$$

In each case, removing the common factors reduces the comparison to one of considering the accuracy of the approximation

$$\tilde{R}_{CO2}(p)\tilde{\Psi}(p) \approx \tilde{R}_{CH4}(p) \qquad AC2.4$$

Because of the commutative and associative properties of such transformations, a transformation of the $CH_4$ source to give an equivalent $CO_2$ source can be described in terms of how well the metric transformation, acting on the $CO_2$ impulse response, reproduces the impulse response for $CH_4$. The application of this relation in the frequency domain (i.e. $p = 2\pi i f$) is noted in the appendix.

*Following this insertion, we propose to use the symbol $\Psi$, with special cases $\Psi_{FEI}$, $\Psi_{GWP}$, $\Psi_{Deriv}$, $\Psi_{Diff}$ and $\Psi_{RM}$, throughout the rest of the section and in the appendix.*

**Line 129** Noted. We propose adding an appendix on a frequency domain analysis. The reason in favour is that Fourier analysis and Fourier transforms are more familiar than Laplace transforms for many scientists. The negatives (which are reasons to use an appendix) are that the analysis is based on complex numbers (as shown in the R code in the supplement) and its formal definition requires limiting processes to ensure convergence of the defining integrals.
**Proposed change:** *Add appendix*

**Line 136** A major reason for considering GWP* and related metrics is because GWP is a poor metric for efficient stabilisation. Nevertheless, a metric that gives perverse behaviour in the short-term is unlikely to gain political acceptance.
**Proposed change:** *Propose adding extra section on practical issues.*

**Line Figs 2,3** These were split for ease of layout in a 2-column journal.
**Proposed change:** *Combine as suggested.*

**Line 164** Propose to rephrase,
**Proposed change:** *Response to reviewer 1 proposes moving this paragraph into new section discussing practicalities. The new section, with rephrasing of line 164, is in a separate post.*

**Conclusions** OK
**Proposed change:** *Proposed re-write posted separately.*

**Line 174** agree.
**Proposed change:** *faster growth rate*
**Line 177** We agree that ratio of airborne fractions is good for all timescales, but approximating this as e-folding rate is not.
**Proposed change:** *....approximating the ratio of airborne fractions as a multiple of the e-folding rate. This approximation can provide a good ....*

**Supplement** Reading the supplement as 'text broken up by code' is, we agree, confusing. It is intended as 'code broken up by text', where the text is inserted in connection with particular parts of the code (i.e. annotated code, as we describe it on line 183). As described, the role of the supplement is to document the code (for review purposes). In the event of acceptance of the paper, we intend to lodge the code in an archive (probably figshare) once we have made any changes as a result of the review process. (We expect that such changes will be confined to the axis rescaling noted in our first post and cosmetic aspects of the graphs.)

---

## Author Comment (AC5) · 6 Jan 2021

**Author comment: Response to referee 1**

**1** The description of Technical Notes is given in the instructions to authors for *Atmospheric Chemistry and Physics* (to which our note was submitted). The apparent lack of a Technical note option for *Earth System Dynamics* seems irrelevant for consideration of our paper.

**2** The Laplace transform has been a standard part of undergraduate STEM education for at least 50 years. It has proved a powerful tool that we would commend to anyone who wishes to **extend** our analysis. Nevertheless, in order to ensure

wider understanding of our analysis we have, at each point, provided alternative ways of describing our results. We also give illustrative examples showing how the various metrics operate in the time domain. We propose to add an appendix (draft to be posted separately) giving a frequency-response interpretation in order to aid communication with a wider audience.

**3** We would agree that the issue of emission equivalence metrics has been "going down a rabbit hole" (in the sense of *Alice in Wonderland*), as shown by the discussion in successive IPCC reports. What our note does is provide a way of comparing some of the recently-proposed metrics in a way that isn't based on the use of specific climate models and/or specific scenarios.

The additions that we propose in response to reviewer 2 (see response regarding section 3.4 in AC4) indicate how our form of analysis might be applicable for considering metrics based on temperature changes.

In general terms, a metric (for $CO_2$-equivalence of $CH_4$) can alternatively be regarded as

- a *statistic* of the $CH_4$ emission history that captures an equivalent $CO_2$ influence on climate;
- an *index*, derived from the $CH_4$ emission history that captures an equivalent $CO_2$ influence on climate;
- a mathematical transformation (which we write as $GWP_0 \times \Psi(p)$) of the methane source $S_{CH4}(t')$ to give an 'equivalent' $CO_2$ source $S_{CO2-eq}(t)$ that generates such an 'index' or 'statistic; — for practical reasons $S_{CO2-eq}(t)$ should depend on $S_{CH4}(t)$ only for earlier emissions, i.e. $t' \leq t$ (see 3.i).

We propose to note this after the additional material noted .

**3.i** We would agree with the reviewer's comment about the impracticality of "providing credits or debits on the assumption that a country or actor will keep its emissions

constant for all times". However none of the metrics that we discuss do this. Credits or debits are based on what actors are doing at the time, in the context of what they have done in the past. We propose to emphasise this characteristic and its importance in the new section on practical implications (see response below on comment on line 165)..

**3.ii** We see the issue of whether nations (or other actors) sign up for unsustainable targets (and then opt out) as distinct from the choice of metrics. This is confirmed by the history of the Kyoto Protocol, with Canada withdrawing and Russia and NZ not taking on second round commitments

**3.iii** On the questions of practicality and effectiveness we see our analysis as a tool from clarifying debate – separate from either side of GWP vs. GWP*. See comment regarding line 165.

**3.iv** At several points, the reviewer notes that our discussion shows the GWP metric doing what it is defined to do. This seems to be missing the point (and says little more than that we appear to have coded our calculations correctly). The point is that the **definition** of GWP leads to a poor specification of equivalence of influences on climate (cf Wigley 1998).

Line by line comments

**Line 64** Source is 5th IPCC assessment. (as noted in code).
**Proposed change:** *reference IPCC, or better still IPCC source*

**Line 66 Proposed change:** *... CO2 contribution, using a GWP of 1, from the oxidation ...*

**Line 73** The analysis is specifically for small perturbations. For larger perturbations, the departures from non-linearity are not just from recent line-by-line calculations

but go back to the analysis by Arrhenius of observations by Langley.
**Proposed change:** *in line 20: with the effect of small perturbations linearised as*

**Line 87** not sure if it is the result or the implications that the reviewer doesn't understand. Assuming the latter, we have expanded our words.
**Proposed change:** *quite close (Enting 2018). Thus in the context of emissions growing with $e$-folding rate, $p$, $GWP_H$ with $H = 1/p$ gives approximate FEI equivalence.*

**Line 123** The parameter $b$ is not dependent on the annual growth rate.
**Proposed change:** *See proposed words in response to comment on section 3.4 by referee 2.*

**Line 137** Need for greater clarity noted.
**Proposed change:** *... long term. This has led to the development of metrics based on rates of change. However, [as discussed in new section below?] for emissions trading on shorter timescales, political acceptance is likely to favour metrics that also have equivalent influences in the short term.*

**Line 145** In part this comment represents aspects of the mis-interpretation that we discuss in detail below in connection with figure 3. With regard to GWP, it is doing what it does, and that in terms of the influence on climate at one particular time, it is a poor specification of equivalence in this case. (see general comment 3.1v above). This is not a new result – we cite Reilly et al, 1999 (and propose to add Wigley 1998) as an example of a study that points out the problems. The point of Figure 3 is that the other metrics do a lot better. The qualitative behaviour of the various cases could be anticipated from the curves in Figure 1, but we think that a specific quantitative example is valuable.
**Proposed change:** *....defining emission equivalence for constant sources.*

**Line 151 Proposed change:** *After $t = 150$ the forcing from equivalence defined by*

*the Cain et al. 2019 metric (dashed line) starts to increase,. This is due to the
contribution that corresponds to 0.25 times GWP when $S_{CH4}(t) \approx S_{CH4}(t-20)$.*

**Line 174** noted, bit see general comment 3.iv above.
   **Proposed change:** *$CH_4$ contribution to radiative forcing ....*

**Line 159** We disagree, and find aspects of the comment incorrect. Targets are com-
   monly set in terms of $CO_2$-equivalent concentrations, related to temperature
   changes by the equilibrium climate sensitivity. $CO_2$ concentration equivalence
   is defined in terms of radiative forcing. We have added words to note the con-
   nection. A 'metric' is a different thing from a 'target'. The reviewer's comments
   about types of metric is irrelevant to our words about targets. The term 'objective
   function' is usually used to denote a function that is minimised in an optimisation
   calculation. Most metrics (with the exception of those that incorporate economic
   aspects) are neither calculated nor defined in that way.
   **Proposed change:** *... radiative forcing targets, commonly expressed as $CO_2$-
   equivalent concentrations, can be ... [This whole paragraph may become part of
   new section on practical aspects]*

**Line 156** We think the Lauder et al. analysis is still valid for the specific case for which
   it was undertaken. However the approach is less suitable for wider application,
   in part because of the forward-looking aspects. The reverse trade would give
   present credits for future promises. Lauder et al also considered only the specific
   case of relating constant $CH_4$ emissions to one-off $CO_2$ emissions and does not
   treat the more general case of non-zero rates of change of $CH_4$ emissions. Thus
   the Lauder result becomes a special case of GWP*.
   **Proposed change:** *None*

**Line 165** We see our note as providing better understanding GWP vs GWP* and similar
   metrics. We are keen to keep this analysis separate from discussions of what

might be politically achievable. We think such deeper analysis needs to be done by others with greater expertise in such areas.

However, we propose a new section that brings together mathematical aspects that may bear on practicality and political acceptability.

**Fig 3** This comment represents a mis-understanding of what GWP is doing.

**a** for a source $S(.)$ the atmospheric content at time t is $M_X(t) = \int_0^t R(t - t')\, S_X(t')\, dt'$

**b** The integrated radiative forcing at time $t$ is $a_X \int_0^t M_X(t')dt' = a_X \int_0^t \int_0^{t'} R(t' - t'')\, S_X(t'')\, dt''dt'$

**c** For a pulse source, $\delta(t)$, relation (a) reduces to $M_X(t) = R_X(t)$ and relation (b) reduces to $a_X \int_0^t M_X(t')dt' = a_X \int_0^t R('t)dt'$

**d** For a unit step source, (a) reduces to $M_X(t) = \int_0^t R(t')\, dt'$

– The fact that the radiative forcing from a unit step (which is, apart for the ramp up, what we are plotting) is the same as the integrated radiative forcing from a unit pulse (which is what defines GWP) reflects the fact that combinations of convolution integrals are commutative and associative. This is an obvious property when using Laplace transforms, but we propose to emphasise it more. Details are given in our response to reviewer 2, regarding section 3.4.

– Thus for the sources of $CO_2$ and $CH_4$ ( scaled by the GWP for time horizon $H$) the radiative forcings from a unit step will be equal at time $H$, and the integrated forcings from a pulse source will also be equal at time $H$. It is the former case that we plot.

---

## Author Comment (AC6) · 11 Jan 2021

Author comment: Proposed appendix

[Note: equation and section numbers refer to the version published in *At. Chem. Phys Discuss*].

The Laplace transform provides a natural formalism for analysing causal initial value systems. However Fourier transforms and Fourier analysis has wide familiarity and can be used to describe our results.

For a periodic variation with exponentially increasing amplitude, equation (5) generalises to

$$\int_{-\infty}^{t} \exp(\alpha t' + i\omega t')R(t-t')\,dt' = \exp(\alpha t + i\omega t)\int_{0}^{\infty} R(t')\exp(-\alpha t' - i\omega t')\,dt'$$

Interactive
comment

For $R_{CO2}$, this relation requires $\alpha > 0$ in order to have the lower limit of the left-hand integral and the upper limit of the right hand integral defined. The $\alpha \to 0$ limit shows the relation between the Laplace transform and the Fourier transform, which, for functions with $R(t) = 0$ for $t < 0$, is given by the integral on the right.

Section 3 [ as modified in response to review comments] noted that metric transformations defined by

$$a_{CO2}\,\tilde{S}_{CO2\text{-}eq}(p) = a_{CH4}\,\tilde{\Psi}(p)\,\tilde{S}_{CH4}(p)$$

can be assessed in radiative forcing terms by the accuracy of the approximation

$$a_{CO2}\,\tilde{R}_{CO2}(p)\,\tilde{S}_{CO2\text{-}eq}(p) = a_{CH4}\,\tilde{R}_{CO2}(p)\,\tilde{\Psi}(p)\,\tilde{S}_{CH4}(p) \approx a_{CH4}\,\tilde{R}_{CH4}(p)\,\tilde{S}_{CH4}(p)$$

which reduces to comparing

$$\tilde{R}_{CO2}(p)\,\tilde{\Psi}(p) \approx \tilde{R}_{CH4}(p)$$

where for FEI equivalence, the approximation becomes exact equality.

A frequency domain interpretation can be obtained by putting $p = 2\pi i f$. In these terms, the metric transformation is acting like a frequency equaliser in an audio system.

The phases of the complex numbers in the relations above capture the phase shifts for the various frequencies. For the present we show only the resulting amplitudes, given by the moduli, $|z|$ of the complex value, and ignore the phase (noting that the modulus of a product is the product of the moduli).

Figure 3 [now that old 3 becomes 2b] sets $p = 2i\pi f$ to evaluate the various cases considered in the paper, as functions of frequency $f$ in cycles per year. It shows
- $|\tilde{R}_{\mathrm{CH4}}(p)|$, the 'target' for FEI equivalence; the zero frequency value is the perturbation lifetime;

- $|\tilde{R}_{\mathrm{CO2}}(p)\tilde{\Psi}_{\mathrm{GWP}}(p)|$, i.e. a multiple of the $CO_2$ response, growing indefinitely as frequency goes to zero;

- $|\tilde{R}_{\mathrm{CO2}}(p)\tilde{\Psi}_{\mathrm{Deriv}}(p)|$ which gives a better approximation over a wider range of frequencies;

- $|\tilde{R}_{\mathrm{CO2}}(p)\tilde{\Psi}_{\mathrm{Diff}}(p)|$ which gives a further improvement, but a notable discrepancy for cycles whose period is near the 20-year interval use in the difference calculation;

- $|\tilde{R}_{\mathrm{CO2}}(p)\tilde{\Psi}_{\mathrm{RM}}(p)|$ which gives a still closer fit of the range of frequencies shown.

Caption for Figure: Frequency response for the various cases of $|\tilde{R}_{\mathrm{CO2}}(p)\,\tilde{\Psi}(p)|$ discussed above, compared to the actual frequency response, $|R_{CH4}(p)|$ to periodic $CH_4$ emissions (solid line), using $p = 2\pi i f$.
* * *
[Figure]

**Fig. 1.** See body of post for caption with Latex formatted,

---

## Author Response (AR1)

**ACP-2020-996: Responses and changes**

I. Enting and N Clisby. Typeset January 28, 2021. File all.tex.

**Context**

This document gives our authors' responses in the form comment/response/change as requested by the editorial office.

The vast majority of the response/change entries are as posted in author comments AC2 and AC3.

**Reviewer 1: General comment 1** I could not find the definitions of the manuscript type technical note on the ESD website (https://www.earth-system-dynamics.net/about/manuscript_types.html), so I cannot comment on the style of the submitted manuscript.

**Author response** The description of Technical Notes is given in the instructions to authors for *Atmospheric Chemistry and Physics* (to which our note was submitted). The apparent lack of a Technical note option for *Earth System Dynamics* seems irrelevant for consideration of our paper.

**Proposed change** None.

————

**Reviewer 1: General comment 2** For the mathematically versed, Enting and Clisby provide some gourmet material, I am sure. I am myself not well versed in Laplace transforms, so I cannot comment on those technical aspects of the manuscript and would hope that the reviewer pool contains a knowledgeable person in that regard.

**Author response** The Laplace transform has been a standard part of undergraduate STEM education for at least 50 years. It has proved a powerful tool that we would commend to anyone who wishes to **extend** our analysis. Nevertheless, in order to ensure wider understanding of our analysis we have, at each point, provided alternative ways of describing our results. We also give illustrative examples showing how the various metrics operate in the time domain.

**Proposed change** We propose to add an appendix (draft posted separately as author comment 6) giving a frequency-response interpretation (i.e. notionally based on Fourier transform rather than Laplace transform) in order to aid communication with a wider audience. This draws in part on material originally posted as 'supplementary information'.

————

**Reviewer 1: General comment 3** Aside from the mathematical beauty of the discussed equivalence metrics: IMHO, this branch of equivalence metrics has gone down a politically completely impractical rabbit hole. Providing credits or debits to a country or actor under the implicit assumption that that country or actor will keep its emissions constant for all times, is as far away from political realities and practicalities as it can be. The reason is that the metric value of a few hundred or a few thousand (the value of lowering the level of CH4 emissions compared to a one-off emission of CO2) would make countries focus 100% on methane reductions (which would negate any need to do anything else, while methane emissions can be lowered) and as soon as methane emissions cannot be lowered, but increase, e.g. as agricultural production increases etc, the country would face such a heavy penalty that it cannot possibly offset that methane increase within a target period (e.g. under the Paris Agreement). Thus, the country would drop out of the international regime. In summary, these emission-rate focused equivalence metrics like GWP*, while solving a scientific, mathematical problem nicely, would make any international effort of reducing a basket of GHGs politically impractical.

**Author response** We would agree that the issue of emission equivalence metrics has been "going down a rabbit hole" (in the sense of *Alice in Wonderland*), as shown by the discussion in successive IPCC reports. What our note does is provide a way of comparing some of the recently-proposed metrics in a way that isn't based on the use of specific climate models and/or specific scenarios.

The additions that we propose in response to reviewer 2 (see response regarding section 3.4 in AC4) indicate how our form of analysis might be applicable for considering metrics based on temperature changes.

In general terms, a metric (for $CO_2$-equivalence of $CH_4$) can alternatively be regarded as

- a *statistic* of the $CH_4$ emission history that captures an equivalent $CO_2$ influence on climate;

- an *index*, derived from the $CH_4$ emission history that captures an equivalent $CO_2$ influence on climate;

- a mathematical transformation (which we write as $\mathsf{GWP}_0 \times \Psi(p)$) of the methane source $S_{CH4}(t')$ to give an 'equivalent' $CO_2$ source $S_{CO2-eq}(t)$ that generates such an 'index' or 'statistic; — for practical reasons $S_{CO2\text{-}eq}(t)$ should depend on $S_{CH4}(t)$ only for earlier emissions, i.e. $t' \leq t$ (see 3.i).

The concept that the $\Delta S_{CH4}$ is being transformed to produce an 'index' or 'statistic' is included in the additional material noted.

**Author response 3.i** We would agree with the reviewer's comment about the impracticality of "providing credits or debits on the assumption that a country or other actor will keep its emissions constant for all times". However none of the metrics that we discuss do this. Credits or debits are based on what actors are doing at the time, in the context of what they have done in the past. We propose to emphasise this characteristic and its importance in the new section on practical implications (see response below on comment on line 165).

**Author response 3.ii** We see the issue of whether nations (or other actors) sign up for unsustainable targets (and then opt out) as distinct from the choice of metrics. This is confirmed by the history of the Kyoto Protocol, with Canada withdrawing and Russia and NZ not taking on second round commitments even though the commitments were defined in terms of GWP.

**Author response 3.iii** On the questions of practicality and effectiveness we see our analysis as a tool for clarifying debate – separate from either side of GWP vs. GWP*. See comment regarding line 165.

**Author response 3.iv** At several points, the reviewer notes that our discussion shows the GWP metric doing what it is defined to do. This seems to be missing the point (and says little more than that we appear to have coded our calculations correctly). The point is that the **definition** of GWP leads to a poor specification of equivalence of influences on climate (cf Wigley 1998).

**Proposed change** None.

————

**Reviewer 1: Line 64** : Provide the reference for the assumed CH4 methane lifetime.
**Author response** Agree.
**Proposed change** Reference IPCC AR5, Chapter 8.

————

**Reviewer 1: Line 66:** Clarify whether the authors suggest adding one additional CO2 contribution, i.e. increasing the metric value of CH4 by 1 or another value

**Author response**  Agree.,
**Proposed change**  .. CO2 contribution, using a GWP of 1, from the oxidation ...

———

**Reviewer 1: Line 73:**  The linearisation of CO2 forcing  by implying $F = \alpha * R$ (the airborne mass) is not necessarily in line with recent line-by-line codes (Etminan et al., 2016). While a valid linearisation for small deviations of emissions, please bring that to the readers attention.
**Author response**  . The analysis is specifically for small perturbations. For larger perturbations, the departures from non-linearity are not just from recent line-by-line calculations but go back to the analysis by Arrhenius of observations by Langley.
**Proposed change** *in line 20*: with the effect of small perturbations linearised as

———

**Reviewer 1: Line 87** :  Apologies, that I do not get this.  More explanation of this line would be useful.
**Author response**  not sure if it is the result or the implications that the reviewer doesn't understand. Assuming the latter, we have expanded our words.
**Proposed change**  ....  quite close (Enting 2018). Thus in the context of emissions $\Delta S_{CH4}$ growing with $e$-folding time, $H$, $\text{GWP}_H$ gives approximate FEI equivalence. Specifically $\text{GWP}_{100}$ gives approximate equivalence for 1% per annum growth rate and, as shown in Figure 1, about a 30% underestimate for the 2% per annum growth rate that approximately characterises 20th century changes.

———

**Reviewer 1: Line 123** :  Why should the parameter b be dependent of an annual growth rate  and what would be the advantage of it matching 100-year GWPs in an exponential growth scenario? Explain. . .
**Author response**  The parameter $b$ is not dependent on the annual growth rate and is not linked to any particular scenario.
**Proposed change** *See proposed words in response to comment on section 3.4 by referee 2 on the mathematical properties mean that scenarios 'factor out' (but note that the equivalent approach was used by Wigley 1998).*

———

**Reviewer 1: Line 137** :  it is unclear to me what the authors mean with "When metrics are used for emissions trading, the behaviour at shorter timescales becomes important". The timescale of interest for a metric is ideally roughly representative with the objective function that policy-makers would like to pursue: like minimising climate change over the next 100 years.  Or to limit peak warming in 40 years from now etc.  Emission trading only operationalises the emphasis that you would place on the mitigation of one GHG over another.  Emission trading itself does not favour shorter timescales in terms of the metrics. The authors should explain / clarify what they mean.
**Author response**  Need for greater clarity noted.
**Proposed change**  ... long term. This has led to the development of metrics based on rates of change. However, [as discussed in new section 5 ] for emissions trading on shorter timescales, political acceptance is likely to favour metrics that also have equivalent influences in the short term.

———

**Reviewer 1: Line 145** :  in this type of context is imprecise. The 100-year GWP does, as Figure 3 sort of suggests, pretty much exactly what it promises to do: Creating emission equivalence in terms of cumulative radiative forcing over the 100-year time horizon.

**Author response** In part this comment represents aspects of the mis-interpretation that we discuss in detail below in connection with figure 3. With regard to GWP, it is doing what it does, and that in terms of the influence on climate at one particular time, it is a poor specification of equivalence in this case. (see general comment 3.1v above). This is not a new result – we cite Reilly et al, 1999 (and propose to add Wigley 1998) as an example of a study that points out the problems. The point of Figure 3 (now fig 2b) is that the other metrics do a lot better. The qualitative behaviour of the various cases could be anticipated from the curves in Figure 1, but we think that a specific quantitative example is valuable.

**Proposed change** ... .defining emission equivalence for constant sources.

——

**Reviewer 1: Line 151** : The sentence The increase after several centuries reflects. . . is unclear. What does it refer to? Please clarify.

**Author response** Noted

**Proposed change** After $t = 150$ the forcing from equivalence defined by the Cain et al. 2019 metric (dashed line) starts to increase,. This is due to the contribution that corresponds to 0.25 times GWP when $S_{CH4}(t) \approx S_{CH4}(t - 20)$

——

**Reviewer 1: Line 174** : It is unclear to me why the authors write that for GWP-H, with H being the time horizon, say H = 100yrs, GWP . . . under-estimates the CH4 contribution from shorter timescales. Well, the GWP-100 is defined that the integral of radiative forcing should be the same from a unit CO2 and a unit CH4 emissions over a 100-year time horizon. So, by design, GWP will then under-estimate the CH4 contribution from timescales that are shorter than the cross-over point ($0 > x > 100$), and afterwards over-estimate the CH4 contribution. If CH4 contribution refers to the radiative forcing. Please either clarify, correct, or both ..

**Author response** Noted, but see general comment 3.iv above.

**Proposed change** $CH_4$ contribution to radiative forcing (in new paragraph in new section 5).

——

**Reviewer 1: Line 159** : The authors write The goal of defining emission equivalence is to allow for emissions of different . . . so that a given radiative forcing target can be achieved for the least economic cost. Well, metrics have various explicit or implicit objective functions, but radiative forcing targets is not usually one. It is either cumulative radiative forcings (GWPs), which approximately the integrated warming over a certain period of time, or it is warming at a particular point in time (GTPs) etc.. Please clarify / rephrase.

**Author response** We disagree, and find aspects of the comment incorrect. Targets are commonly set in terms of $CO_2$-equivalent concentrations, related to temperature changes by the equilibrium climate sensitivity. $CO_2$ concentration equivalence is defined in terms of radiative forcing. We have added words to note the connection. A 'metric' is a different thing from a 'target'. The reviewer's comments about types of metric is irrelevant to our words about targets. The term 'objective function' is usually used to denote a function that is minimised in an optimisation calculation. Most metrics (with the exception of those that incorporate economic aspects) are neither calculated nor defined in that way.

**Proposed change** .. radiative forcing targets, commonly expressed as $CO_2$-equivalent concentrations, can be ... [This whole paragraph is incorporated in new section 5 on practical aspects]

——

**Reviewer 1: Line 156** : It is unclear what the authors want to say with the sentence For a specific case, Lauder et al. (2013) suggested an approximate equivalence to changes in methane emissions balanced by an ongoing future CO2 uptake from growing trees. It is unclear whether the authors support that conclusion. In the light of the above discussion, with zero-CO2 emissions being equivalent with non-changing CH4 emission rates (after a CH4 concentrations reach their new equilibrium), whether the Lauder et al. conclusion (as presented here) still holds. Please clarify.

**Author response** We think the Lauder et al. analysis is still valid for the specific case for which it was undertaken. However the approach is less suitable for wider application, in part because of the forward-looking aspects. The reverse trade would give present credits for future promises. Lauder et al also considered only the specific case of relating constant $CH_4$ emissions to one-off $CO_2$ emissions and does not treat the more general case of non-zero rates of change of $CH_4$ emissions. Thus the Lauder result becomes a special case of GWP*.

**Proposed change** None.

――――

**Reviewer 1: Line 165** : In the conclusion, I'd appreciate a bit of discussion from the authors on the general point I raise above, i.e. the practicality of rate-based emission metrics vs GWP-100 in a real-world context

**Author response** We see our note as providing better understanding GWP vs GWP* and similar metrics. We are keen to keep this analysis separate from discussions of what might be politically achievable. We think such deeper analysis needs to be done by others with greater expertise is such areas.

However, we propose a new section that brings together mathematical aspects that may bear on practicality and political acceptability.

**Proposed change** New section 5.

――――

**Reviewer 1: Figure 3** : Maybe I misunderstand the scaling of the y-axes, but the dashed GWP line should not cross the solid CH4 line in year 100, but earlier. The GWP-equivalence is given if the cumulative radiative forcing over a 100-year time horizon is equal. Thus, the crossing of the dashed line should be such that the integral underneath the solid and the dashed (GWP) line is identical from year 50 to year 150. Please clarify or explain why I am wrong in thinking that.

**Author response** This comment represents a mis-understanding of what GWP is doing.

a for a source $S(.)$ the atmospheric content at time t is $M_X(t) = \int_0^t R(t - t') \, S_X(t') \, dt'$

b The integrated radiative forcing at time $t$ is
$a_X \int_0^t M_X(t') dt' = a_X \int_0^t \int_0^{t'} R(t' - t'') \, S_X(t'') \, dt'' dt'$

c For a pulse source, $\delta(t)$, relation (a) reduces to $M_X(t) = R_X(t)$ and relation (b) reduces to $a_X \int_0^t M_X(t') dt' = a_X \int_0^t R('t) dt'$

d For a unit step source, (a) reduces to $M_X(t) = \int_0^t R(t') \, dt'$

– The fact that the radiative forcing from a unit step (which is, apart for the ramp up, what we are plotting) is the same as the integrated radiative forcing from a unit pulse (which is what defines GWP) reflects the fact that combinations of convolution integrals are commutative and associative. This is an obvious property when using Laplace transforms, but we propose to emphasise it more because it is these properties that make it possible to 'factor out' the dependence on emission scenarios. Details are given in our response to reviewer 2, regarding section 3.4.

- Thus for the sources of $CO_2$ and $CH_4$ (scaled by the GWP for time horizon $H$) the radiative forcings from a unit step will be equal at time $H$, and the integrated forcings from a pulse source will also be equal at time $H$. It is the former case that we plot.

**Proposed change**  No change at this point, but emphasise the commutative and associative properties that underlie the discussion above — see response to reviewer 2 regarding section 3.4. Note that figure 3 is now 2b in response to reviewer 2.

———

**Reviewer 2: General comment**  *We have broken this general comment into separate points in order to help match our response to the various points in this comment*

This technical note provides an interesting theoretical analysis of CO2 forcing equivalence. Even though the note is designed to be mainly technical it poses many issues that require more discussion here. The analysis provides a nice comparison of different metrics using a Laplace framework.

**i** It seems that the forcing equivalent CO2 emissions can be expressed as a Reduced Model in equation (20). It should be explained if this is a simpler methodology than inverting the CO2 response function as in Wigley 1998.

**ii** It is not clear from this paper how this could be applied in any policy context. Typically emission metrics are presented and used as a single number (or two numbers, short and long, in Levasseur et al. papers). The authors should explain in what context a continuous function $\Delta S_{CO2}(t)$ could be useful as a metric.

**iii** It appears (like Cain et al. 2019) that the metric depends on the past emission history. This means that the larger the past emissions the lower the metric. This could be controversial politically and at least some short discussion is warranted on how/why past behaviour influences the future, and what the implications might be for policy.

**iv** Given the similarities of equations 15 and 20, it would be useful to compare them more fully. Do the two terms on the left hand side of equation 20 correspond to the two terms on the right hand side of equation 15? Can the 4 and 3.75 coefficients in equation 15 be related to the b coefficient in equation 20?

**v** The conclusions would be better as flowing text, rather than the series of bullet-like points.

**Author response** .

**i** The formal inversion of the FEI relation can be done exactly (if the responses are expressed as sums of exponentials) but the resulting inversion relation would be impractical as a metric because (a) the large number of parameters required (b) the likely ill-conditioning involved in determining the coefficients in $R_{CO2}$ (as in the classic example given by Lanczos)

**ii** We are not proposing a continuous metric. $\Delta S_{CO2}(t)$ is not a metric. It is the result, for a particular time $t$, of applying a metric. The metric is the process of going from $\Delta S_{CH4}(.)$ to the CO2-equivalent. In general mathematical terms this would be a functional. Restricting such functionals to time-invariant linear operators whose Laplace transforms are rational functions restricts consideration to metrics defined as linear integro-differential operators. The full inversion of the Wigley FEI relation can be expressed in this way (most easily by using Laplace transforms) if the $CH_4$ and $CO_2$ responses are sums of exponentials. However, our analysis suggests that useful approximations can be obtained using much simpler expressions.

The various metric processes that we consider for generating $\Delta S_{CO2}(t)$, each applicable at any single time $t$, are

- multiply $\Delta S_{\mathrm{CH4}}(t)$ by a constant (ie, GWP approach);
- multiply $\frac{d}{dt}\Delta S_{\mathrm{CH4}}(t)$ by a constant — in practice this would require a specification of how the derivative is defined;
- combine current $\Delta S_{\mathrm{CH4}}(t)$, with the 20-year difference $4\Delta S_{\mathrm{CH4}}(t) - 3.75\Delta S_{\mathrm{CH4}}(t-20)$;
- take current $\Delta S_{\mathrm{CH4}}(t)$ offset by weighted integral over past emission perturbations.

**iii** Any metric except GWP is going to depend on multiple times, although the derivative metrics leave the process for doing so undefined.

**iv**. As they stand, equations (15) and (20) include the emission scenarios. Once these are factored out, the comparison reduces to comparing $\tilde{\Psi}_{\mathrm{Diff}}(p)$ to $\tilde{\Psi}_{\mathrm{RM}}(p)$ . This comparison is done in Figure 1.

**v** Agreed.

**Proposed change** :

**i** Complexity of full inversion of FEI noted. .

**ii** Emphasise that it is the transformation that is the metric, i.e. not the **result** of applying the metric.

**iii** None.

**iv** None.

**v** Conclusions rewritten as continuous text. See response on **Conclusions** below. Draft posted as author comment.

————

**Reviewer 2: Line 16** : I'm not sure "so-called" is a necessary qualifier for greenhouse gases.

**Author response** the use of 'so-called' captures the fact that actual greenhouses don't work by changing radiation balance. Our bending over backwards for correctness reflects the politicisation of climate science, particularly in Australia and the USA.

**Proposed change** Removed, but leave final decision to editor.

————

**Reviewer 2: Line 21** : $a_X$ needs to be defined.

**Author response** Noted

**Proposed change** where $a_X$ is the radiative efficiency in mass units: the amount of change in radiative forcing per unit mass increase for constituent $X$ in the atmosphere.

————

**Reviewer 2: Sections 3.2 and 3.3** need to refer to figure 1 and it needs to be clearer which lines in the figure are being referred to in these sections.

**Author response** Agreed

**Proposed change** insert as shown by the *** line,' after lines 82, 93, 104, 114. Similar change also made in Section 4.

————

**Reviewer 2: Section 3.4** : The key parameter here is $b$ so there needs to be more explanation of what this might relate to physically. The text explains a derivation for a 1%/yr growth rate. Would $b$ be completely different for a different emission profile (e.g. figure 2)?

**Author response** We regard the parameter $b$ as being an empirical fit that has no specific physical meaning. The reduced model is fitting the ratio of two response functions whose parameters are themselves empirical fits whose parameters have only distant connection to the underlying processes involved. However, the important point is that $b$ is independent of the growth rate used in the example.

**Proposed change** *Propose inserting, after line 65:*

A general linear, time-invariant equivalence relation defined by

$$a_{CO2}\Delta\tilde{S}_{\text{CO2-eq}}(p) = a_{\text{CH4}}\tilde{\Psi}(p)\Delta\tilde{S}_{\text{CH4}}(p) \qquad\qquad AC2.1$$

In the time domain, such a metric can be regarded as a process that extracts, from the history of $CH_4$ emissions, an 'index' or 'statistic' that gives a $CO_2$ equivalence. Such a metric can be assessed in radiative forcing terms by the accuracy of the approximation

$$a_{\text{CO2}}\tilde{R}_{\text{CO2}}(p)\Delta\tilde{S}_{\text{CO2-eq}}(p) = a_{\text{CH4}}\tilde{R}_{\text{CO2}}(p)\tilde{\Psi}(p)\Delta\tilde{S}_{\text{CH4}}(p) \approx a_{\text{CH4}}\tilde{R}_{\text{CH4}}(p)\Delta\tilde{S}_{\text{CH4}}(p) \quad AC2.2$$

If the global temperature response is linearised using a response function $U(t)$, as in done for example in AR5-WG1-Ch8, then equivalence in temperature perturbations can be analysed in terms of the approximation

$$\tilde{U}(p)a_{\text{CO2}}\tilde{R}_{\text{CO2}}(p)\Delta\tilde{S}_{\text{CO2-eq}}(p) = \tilde{U}(p)a_{CH4}\tilde{R}_{CO2}(p)\tilde{\Psi}(p)\Delta\tilde{S}_{CH4}(p)$$

$$\approx \tilde{U}(p)a_{CH4}\tilde{R}_{CH4}(p)\Delta\tilde{S}_{CH4}(p) \qquad\qquad AC2.3$$

In each case, removing the common factors reduces the comparison to one of considering the accuracy of the approximation

$$\tilde{R}_{CO2}(p)\tilde{\Psi}(p) \approx \tilde{R}_{CH4}(p) \qquad\qquad AC2.4$$

As Wigley (1998) noted 'If $CO_2$-equivalence is based on radiative forcing, and calculated accurately for non-$CO_2$ gases, then the temperature and sea-level implications of the [Kyoto] Protocol may be calculated from the $CO_2$-alone case'.

Because of the commutative and associative properties of such transformations, a transformation of the $CH_4$ source to give an equivalent $CO_2$ source can be described in terms of how well the metric transformation, acting on the $CO_2$ impulse response, reproduces the impulse response for $CH_4$. The application of this relation in the frequency domain (i.e. $p = 2\pi i f$) is noted in the appendix.

*Note that the quote from Wigley is an addition to what we posted in the on-line comments. Following this insertion, we propose to use the symbol $\Psi$, with special cases $\Psi_{FEI}$, $\Psi_{GWP}$, $\Psi_{Deriv}$, $\Psi_{Diff}$ and $\Psi_{RM}$, throughout the rest of the section and in the appendix.*
* * *
**Reviewer 2: Line 129** : This sentence about frequency aliasing is too cryptic as written here. This either needs to be expanded or removed.

**Author response** Noted. We propose adding an appendix on a frequency domain analysis. The reason in favour is that Fourier analysis and Fourier transforms are more familiar than Laplace transforms for many scientists. The negatives (which are reasons to use an appendix) are that the analysis is based on complex numbers (as shown in the R code in the supplement) and its formal definition requires limiting processes to ensure convergence of some of the defining integrals.

**Proposed change** Add appendix, with note on aliasing.
* * *
**Reviewer 2: Line 136** : Why does the behaviour at shorter timescales become important when used for emission trading?

**Author response** A major reason for considering GWP* and related metrics is because GWP is a poor metric for efficient stabilisation. Nevertheless, a metric that gives perverse behaviour in the short-term is unlikely to gain political acceptance.

**Proposed change** adding new section 5 on practical issues

——

**Reviewer 2: Figures 2 and 3** : Since these need to be viewed together I suggest combining into two panels (a) and (b) of figure 2.

**Author response** These were split for ease of layout in a 2-column journal.

**Proposed change** Combined as suggested.

——

**Reviewer 2: Line 164** : Some explanation is needed why least cost overshoots the radiative forcing target for GWP100.

**Author response** Response to reviewer 1 proposes moving this paragraph into new section discussing practicalities. The new section, with rephrasing of line 164, is in a separate post.

**Proposed change** New section.

——

**Reviewer 2: Conclusions** : I found this bullet style very difficult to read or to pull out the key points. I suggest rewriting completely and focusing on the key points as to what has been concluded, and what should a reader take away.

**Author response** OK.

**Proposed change** Revised conclusions posted separately.

——

**Reviewer 2: Line 174** : It might be better to write as"faster growth rate", since "shorter timescales" might be confused with GWP20 etc., and line 177. Also I'm not sure this is very policy relevant as no plausible future emission scenario has exponentially growing methane emissions.

**Author response** agree on terminology

**Proposed change** faster growth rate

**Author response** The comment about exponentially growing emissions seems irrelevant for two reasons: (i) our analysis is about perturbations and so an exponential growth from leakage when replacing coal by gas is possible (and may occur, for example, for Australian emissions); (ii) as with Fourier transforms, the Laplace transform is based on representing functions of time as a linear combination of specified functions, where some of the weightings may be negative. The use of Laplace transforms is not restricted to looking at one exponential growth at a time any more than Fourier analysis is restricted to analysing one frequency at a time.

**Author response** where we refer to exponentially growing emissions (line 109 as submitted) we insert $\Delta S_{CH4}$ to emphasise that we are talking about perturbations.

——

**Reviewer 2: 177** : I think shorter timescales here means something different to line 174. It is not obvious why the ratio of airborne fractions is a good representation of long-term behaviour, but not for emissions trading.

**Author response** We agree that ratio of airborne fractions is good for all timescales, but approximating this as e-folding rate is not.

**Proposed change** ...approximating the ratio of airborne fractions as a multiple of the e-folding rate. This approximation can provide a good ....

——

**Reviewer 2: Supplementary material** The text and code need to be separated here. It was very difficult to follow the text when it was so broken up by code.

**Author response** Reading the supplement as 'text broken up by code' is, we agree, confusing. It is intended as 'code broken up by text', where the text is inserted in connection with particular parts of the code (i.e. annotated code, as we describe it on line 183). As described, the role of the supplement is to document the code (for review purposes). In the event of acceptance of the paper, we intend to lodge the code in an archive (probably figshare) once we have made any changes as a result of the review process. (We expect that such changes to the code will be confined to the axis rescaling noted in our first post (AC1) and cosmetic aspects of the graphs.)

**Proposed change** New appendix including those parts of supplementary information relevant to frequency response , independent archiving of updated code. , No further use of supplementary information. .

————

**Other changes** Rescale graph axes so that GWP includes indirect effects, citing IPCC AR5